# Potential of Advanced Consolidants for the Application on Sandstone

**Monika Remzova [1], Radek Zouzelka [1] , Jaroslav Lukes [2] and Jiri Rathousky [1,*]**

[1]  J. Heyrovsky Institute of Physical Chemistry of the CAS, Dolejskova 3, 18223 Prague, Czech Republic; monika.remzova@jh-inst.cas.cz (M.R.); radek.zouzelka@jh-inst.cas.cz (R.Z.)

[2]  Czech Technical University in Prague, Faculty of Mechanical Engineering, Technicka 4, 166 07 Prague, Czech Republic; jaroslav.lukes@fs.cvut.cz

*  Correspondence: jiri.rathousky@jh-inst.cas.cz

**Abstract:** Alkoxy-based consolidants are widely used for conservation of cultural heritage objects made from sandstones. Even if such consolidants were introduced into practice already in the 19th century, their performance has been enhanced by several modifications to their composition only recently. To obtain a relevant comparison of their application potential, both commercial (Remmers KSE OH and SurfaPore FX SB) and a self-developed particle-modified ethyl silicate consolidant supplemented with two phosphate-based ones, were assessed. Importantly, the potential toxicity of our novel consolidants was considered. Since the stone substrate should mimic the properties of naturally weathered stone, sandstone from the Msene quarry in Central Bohemia, characterized by a high porosity and relatively low mechanical strength, is selected. From practical point of view, the long-term durability of the consolidation effect is crucial compared to the initial level of consolidation. Regarding the determination of durability of the sandstone consolidation according to an American Society for Testing and Materials (ASTM) accelerated weathering test, we performed mechanical tests in micro- (nanoindentation) and macroscale (drilling resistance measurements). The cohesion of the consolidant xerogels in the pores were determined by sorption experiments in gas phase. The durability of our $TiO_2$ and $ZnO$ particle-modified consolidants is superior to that of the commercial products. The aqueous diammonium hydrogen phosphate-based consolidant, which also shows exceptional durability, reveals itself to be a promising product for not only carbonate but also sandstone materials.

**Keywords:** consolidation; sandstone; alkoxide consolidants; diammonium hydrogen phosphate; calcium oxalate; accelerated weathering; durability

## 1. Introduction

Alkoxysilanes have been the most widely used stone consolidants since the 19th century [1]. Two compounds have been dominant: methytrimethoxysilane (MTMOS) and tetraethoxysilane (TEOS). A number of products based on TEOS and its oligomers are available on the market. Their advantages are well known: their versatility, chemical stability, low viscosity, the ability to form silicon–oxygen–silicon (Si–O–Si) bonds [2], harmlessness of the compounds (e.g., water, ethanol) formed during gelation, avoidance of over-consolidation of the stone surface and, last but not least, cost effectiveness and ease of use. Silicon–oxygen bonds are contained in many minerals, thereby contributing to the compatibility between stone and TEOS-based alkoxysilanes. Owing to the advantages listed above, such consolidants easily penetrate deeply into the stone's porous system [3]. They can replace an original binder lost during weathering and, thereby, significantly enhance the cohesion of the stone grains and, thus, overall structure [1].

However, there are serious drawbacks concerning the application of TEOS-based consolidants. Specifically, the resulting silica gel shrinks and tends to develop cracks inside the stone due to the stress caused by increasing capillary force during aging and drying [1,3]. This particular situation leads to the formation of brittle gel fragments that create a secondary capillary network in the stone, which can accelerate the rate of deterioration. Since the xerogel formed by the gelation of TEOS and oligomeric ethyl silicates are compact with only very narrow micropores (1–2 nm in width), the water vapor permeability is substantially hindered, which represents a major danger for the stability of the treated stone. Furthermore, the physico-chemical and structure properties of these consolidant xerogels differ from those of natural sandstones, which consist of quartz particles and an inorganic binder.

This issue has been of major research interest in stone consolidation applications over the past twenty years. There are several approaches to overcoming the above-mentioned shortcomings. Blanco et al. showed that by adding drying control chemical additives (DCCA), such as formamide, the surface tension of the sol was changed and, thus, the evaporation rate was reduced, leading to the suppression of crack formation [4]. Mosquera et al. also reduced cracking by the addition of suitable surfactants (alkylamines), which decreased capillary pressure and surface tension [5]. Another strategy involves the incorporation of elastic segments, such as hydroxyl-terminated polydimethylsiloxane (PDMS), into the xerogel structure; these segments effectively removed the stress caused by capillary pressure [5–9]. Another more general approach is based on the incorporation of various oxide nanoparticles, such as $SiO_2$, $TiO_2$, $Al_2O_3$ and ZnO (so-called particle-modified consolidants, PMC) [3,8,10–15]. Due to the increase in the pore size of PMC-based xerogel and the improvement of its mechanical properties, cracking was reduced, and the consolidation effect enhanced. However, while they have shown significant improvements, these 'new' formulations exhibit considerable limitations due to the complexity of the treated stone objects.

During the development of any novel consolidant, the selection of a suitable stone substrate is fundamental. Theoretically, three main substrates can be used: (i) a naturally aged substrate, (ii) an artificially aged substrate, and (iii) stone as quarried [15]. The first option provides the advantage of weathering under real world conditions and potentially reliable technological transfer into conservation practice. However, there are issues, namely the insufficient availability of stone samples for extensive laboratory testing and the natural inhomogeneity. The modification of stone by artificial weathering (second option) is rather more problematic as the aging processes differ considerably from those in nature. The oft-used thermal treatment, especially of sandstone, causes physico-chemical changes, such as binder hardening, that render such treated materials unsuitable for consolidation tests. The freeze–thaw cycles are unsuitable for large-scale testing. Additionally, salt crystallization and/or acid attack make it difficult to interpret the consolidation performance. Concerning quarried stone (third option), the test material must be very carefully selected so that it accurately mimics the real weathered one.

The Czech Republic is particularly suitable in this respect because it is rich in sandstone quarries, meaning that a wide range of samples differing in texture and mechanical properties are available. Our previous studies have shown that sandstone from the Msene quarry in Central Bohemia is suitable for the evaluation of consolidation performance because it is characterized by high-porosity and relatively low mechanical strength [11,13,14]. This sandstone is characterized by high-purity (over 95% of quartz and a small amount of binder), homogeneity and a well-defined monomodal porous system with a narrow pore-size distribution. Moreover, this stone was used as a construction material for extremely important historical monuments in Prague, such as St. Vitus's Cathedral, Charles Bridge, the Old Town Hall and several government buildings.

The third fundamental aspect, besides the selection of stone substrate to be tested and the choice of consolidant, is the methodology as to how the success of consolidation will be evaluated. From the practical point of view, the long-term durability of the consolidation effect is more pivotal than the initial level of consolidation. When there is a fast loss of the consolidation effect, such consolidation technology would not be suitable for application.

Therefore, in this study, we focus on the aspect of durability and use it as a criterion for the assessment of the consolidation outcome. Specifically, we consolidate Czech-quarried sandstone by various consolidant formulations, including two TEOS-based commercial products (Remmers KSE OH, SurfaPore FX SB), a self-developed TEOS-based PMC consolidant [11], and two self-developed phosphate-based consolidants. Afterwards, we artificially weather the treated samples and determinethe durability of each type of sandstone consolidation by performing mechanical tests on a micro- (nanoindentation) and macroscale (drilling resistance measurements). To evaluate the cohesion of the consolidant xerogels, both before and after incorporation into the sandstone pores, sorption experiments are carried out in the gas phase.

## 2. Materials and Methods

### 2.1. Materials

Concerning the preparation of consolidants, five commercially available metal oxide nanoparticles were used, including Aeroxide® $TiO_2$ P25 (Evonik Industries, Essen, Germany), Aerosil® $SiO_2$ 200 Pharma, Aerosil® $SiO_2$ R805, Aerosil® $SiO_2$ R9200 (all Evonik Industries, Essen, Germany) and NanoZnO (Bochemie, Bohumin, Czech Republic). Detailed characteristics of the nanoparticles are provided in reference [11]. Dynasylan 40 was obtained from Evonik Industries (Essen, Germany), isopropanol (IPA), dibutyltin dilaurate (DBTL), n-octylamine, were all from Sigma–Aldrich (St. Louis, MO, USA), and $Ca(OH)_2$ (Penta, Prague, Czech Republic).

### 2.2. Preparation of the Nanoparticle-Modified Consolidants

The self-developed alkoxide-based consolidants were prepared by adding the nanoparticles (3 wt.%) to Dynasylan®40 [11,13,14]. Afterwards, catalyst n-octylamine (0.18 wt.%) (Alfa Aesar) was added. The mixture obtained was diluted with isopropanol at the ratio of 1:1. Concerning the consolidant containing ZnO nanoparticles, a mixture of n-octylamine (0.18 wt.%) and dibutyltin dilaurate (1 wt.%) was used because n-octylamine was not sufficient to achieve a formation of the gel. An overview of the consolidants is given in Table 1, wherein the designation of Samples, the type of nanoparticles and catalysts used are given, for example, SiGel-SiO$_2$-methyl-oa means that the consolidant contains methylated $SiO_2$ nanoparticles and n-octylamine catalyst.

**Table 1.** Overview of the composition of self-prepared and commercial consolidants.

| Source | Sample | Consolidating Agent | Solvent | Catalyst | Additive |
|---|---|---|---|---|---|
| IPC [1] | SiGel-d | 50% oligomeric TEOS | 50% IPA [2] | dibutyltin dilaurate | - |
| IPC | SiGel $SiO_2$-d | 50% oligomeric TEOS | 50% IPA | dibutyltin dilaurate | $SiO_2$ |
| IPC | SiGel $SiO_2$-octyl-d | 50% oligomeric TEOS | 50% IPA | dibutyltin dilaurate | $SiO_2$-octyl |
| IPC | SiGel-oa | 50% oligomeric TEOS | 50% IPA | n-octylamine | - |
| IPC | SiGel $SiO_2$-oa | 50% oligomeric TEOS | 50% IPA | n-octylamine | $SiO_2$ |
| IPC | SiGel $SiO_2$-octyl-oa | 50% oligomeric TEOS | 50% IPA | n-octylamine | $SiO_2$-octyl |
| IPC | SiGel $SiO_2$-methyl-oa | 50% oligomeric TEOS | 50% IPA | n-octylamine | $SiO_2$-methyl |
| IPC | SiGel $TiO_2$-oa | 50% oligomeric TEOS | 50% IPA | n-octylamine | $TiO_2$ |
| IPC | SiGel ZnO-oa/d | 50% oligomeric TEOS | 50% IPA | n-octylamine + dibutyltin dilaurate | ZnO |
| IPC | HAP | 23% DAP | 77% $H_2O$ | - | $Ca(OH)_2$ |
| Remmers | KSE OH | 75% oligomeric TEOS | 25% MEK | dibutyltin dilaurate | - |
| NanoPhos SA | SP FX | 20% TEOS | 80% IPA | n-octylamine | $CaC_2O_4$ |

[1] IPC = J. Heyrovsky Institute of Physical Chemistry; [2] IPA = isopropyl alcohol; [3] DAP = diammonium hydrogen phosphate.

The second type of self-developed consolidants included hydroxyapatite-based ones. A 2.0 M aqueous solution of diammonium hydrogen phosphate (DAP) was prepared by dissolving the salts in deionized water. This concentration (close to saturation at room temperature) was selected because the higher the DAP concentration, the more calcium phosphate phases are formed by the reaction with the calcium source.

The commercial consolidant KSE OH (Remmers) was used as reference. Compared to our developed consolidants, KSE OH contains only 25% solvents, while ours contains 50%. According to the data sheet, this product should be suitable for the consolidation of weathered, friable natural stones, particularly sandstones, cast stone, renders and mortar. Another commercial consolidant SurfaPore FX WB was obtained from Nanophos SA (Lavrio, Greece). This innovative nano-material is suitable for enhancing the mechanical properties of worn building surfaces [16,17]. To test, the consolidant xerogels aged for two months were used.

The consolidants were applied to stone by capillary suction; the stone specimens were partially immersed (3 mm depth) in the liquid products for 3 h. The application time was determined by measuring the evolution of the capillary fringe (mm) on the lateral surface of the specimens. We concluded that 3 h was the time necessary for the wet fringe of the majority of the consolidants to reach the top of the specimens (3 cm). The samples were weighed before and immediately after the application of the consolidant to calculate the uptake. Additionally, they were re-weighed after complete drying (months after the consolidant application) to calculate the dry yield. During the treatment and curing, the conditions were controlled by a climatic chamber at 25 °C and RH of 75%. The experiments were done in triplicate.

## 2.3. Methods

Textural properties of xerogels and rocks dried at 60 °C were determined by the analysis of adsorption isotherms of nitrogen or krypton at the boiling point of liquid nitrogen (ca 77 K). Prior to the adsorption experiment, the samples were outgassed at 60 °C for 24 h to ensure the complete cleaning of the surface. The experiments were carried out using a Micrometrics 3Flex volumetric adsorption unit. Due to the complex character of the samples' porosity, the obtained isotherms were analyzed by a combination of several methods, especially the Broekhoff–de Boer t-plot and several variants of the Non-Local Density Function Theory (NLDFT). The pore width is described using the IUPAC nomenclature, micropores, mesopores and macropores corresponding to a width of less than 2 nm, 2–50 nm and more than 50 nm, respectively.

The pore size distribution and the total open porosity were determined by mercury intrusion porosimetry using the Quantachrome Poremaster PM-60-13 instrument, working in the pressure range corresponding to pore sizes from several nanometers to about 200 μm. The mercury parameters were as follows: 480 erg cm$^{-2}$ for the surface tension and 140° for the contact angle. Sample size was about 1 cm$^3$.

During the measurement of the water uptake by capillary suction, the stone specimens were partially immersed (3 mm depth) in water for 3 h and the amount of soaked water was determined by weighing [18].

The mechanical properties during the microscale level were tested using an Hysitron TI 750 L Ubi nanoindentation instrument. A three-sided pyramidal diamond Berkovich indenter and the Oliver–Pharr method were used to get the elastic modulus and hardness. Each sample was scanned by in-situ Scanning Probe Microscopy (SPM) imaging to select a suitable place for experimental testing with minimum roughness. The maximum load was $P_{max}$ = 1.0 mN and the load was increased linearly for 5 s to reach maximum, which was held for 2 s and then full unloading was achieved within 5 s. The samples to be measured were embedded in resin, after 24 h they were ground with SiC paper (up to grit #2000) and polished. Finally, the surfaces were cleaned by sonication. The distance between the indents and the total test area were 1 μm and 25 μm$^2$, respectively. Due to the embedding in resin and porosity of samples, single indents were performed to distinguish the resin from the gel in



pores. The location of single indents was selected with the aid of an optical microscope and, for each sandstone sample, at least 30 indents were created.

The mechanical properties during the macroscale level were determined by a Drilling Resistant Measuring System (DRMS, from SINT Technology, Calenzano, Italy) [19]. The cube blocks of sandstone, 3 cm in size, were drilled with drill bits of 4.8 mm in diameter at a rotation speed of 300 rpm and a penetration rate of 30 mm min$^{-1}$.

### 2.4. Accelerated Weathering

The tests were carried out using a Q-Sun Xenon Test Chamber, which reproduces the full spectrum of sunlight (ultraviolet, visible light and infrared), controls the humidity and temperature within the test chamber and simulates rain. Sandstone samples both untreated and treated with all consolidants included were weathered by 50 cycles according to American Society for Testing and Materials (ASTM) G 155-05a Standard Practice for Operating Xenon Arc Light Apparatus for Exposure of Non-Metallic Materials (2005) [20]. This test includes irradiation with xenon light at an increased temperature of 47 °C combined with water spray for one hour, as summarized in Table 2.

**Table 2.** Parameters of the accelerated weathering testing.

| Steps | Exposure Period/ min | Irradiance [1]/ W m$^{-2}$ | Black Standard Temperature/ °C | Chamber Temperature/ °C | Relative Humidity/ % |
|---|---|---|---|---|---|
| 1. light | 40 | 0.55 | 70 ± 2 | 47 ± 2 | 50 ± 5 |
| 2. light and spray | 20 | 0.55 | 70 ± 2 | 47 ± 2 | 50 ± 5 |
| 3. light | 60 | 0.55 | 70 ± 2 | 47 ± 2 | 50 ± 5 |
| 4. dark and spray | 60 | 0.00 | 38 ± 2 | 38 ± 2 | 95 ± 5 |

[1] Wavelength of 340 nm.

## 3. Results and Discussion

### 3.1. Characterization of the Used Rock Substrate

The sandstone used comes from quarries in Msene, central Bohemia (Czech Republic). It is a white-greyish fine-grained sedimentary psammitic rock, dominantly containing quartz (95%), with muscovite and feldspar clasts as accessories. The matrix is formed by clay minerals (kaolinite, chlorite, illite), but its content is very low.

Regarding this sandstone, the main petrophysical properties expressed as average values are porosity (28%), bulk density (1.89 g cm$^{-3}$), and pore volume (0.14 cm$^3$ g$^{-1}$). The narrow monomodal pore size distribution was centered at 30 μm, practically all pores were ranging from 20 to 40 μm, and water absorption (12 wt.%) Further, our adsorption experiments, in which both nitrogen and krypton were used as sorbates (at the boiling point of liquid nitrogen), produced similar surface areas of ca. 0.5 m$^2$ g$^{-1}$, which did not significantly differ from that determined by mercury intrusion porosimetry.

The mechanical performance of the sandstone was determined by three different methods: drilling resistance (9.7 N), compressive strength (28.0 MPa) and three point bending strength (1.3 MPa) measurements. These low mechanical properties confirmed the suitability of our porous material for use in studying the consolidation of deteriorated sandstone.

### 3.2. Characterization of the Consolidant Xerogels

3.2.1. Effect of Catalyst and Added Nanoparticles on Xerogel Texture Properties

A sol–gel process was used to transform liquid consolidants into solid xerogels, whose properties were determined by an extensive adsorption study, the results of which are summarized in Table 3.

**Table 3.** Textural properties of the xerogels determined by sorption measurements.

| Xerogel | Micropore Volume/ cm$^3$ g$^{-1}$ | Mesopore Volume/ cm$^3$ g$^{-1}$ | BET Surface Area [1]/ m$^2$ g$^{-1}$ | Pore Width/ nm | Total Porosity [2]/ % |
|---|---|---|---|---|---|
| SiGel-d | 0.15 | 0.00 | * | 0.8 | 25 |
| SiGel-oa | 0.00 | 0.23 | 198 | 5.0 | 34 |
| SiGel-SiO$_2$-oa | 0.00 | 0.24 | 187 | 6.6 | 35 |
| SiGel-SiO$_2$-methyl-oa | 0.00 | 0.26 | 223 | 5.6 | 39 |
| SiGel-SiO$_2$-octyl-oa | 0.00 | 0.24 | 209 | 5.4 | 35 |
| SiGel-TiO$_2$-oa | 0.00 | 0.22 | 207 | 5.0 | 33 |
| SiGel-ZnO-oa/d | 0.14 | 0.17 | 55 | 1.9; 4–5 | 41 |
| KSE OH | 0.06 | 0.27 | 379 | 1.9; 4–5 | 38 |
| SP FX | 0.00 | 0.01 | 12 | 2–4 | 2 |

[1] BET equation is not applicable in narrow micropores. [2] Total porosity was calculated from the total pore volume.

The addition of nanoparticles to the sols whose gelation was catalyzed by dibutyltin dilaurate did not lead to the formation of mesoporosity in corresponding xerogels. This might be due to formation of a narrow corona around the nanoparticles, the corona itself being identified as a micropore. Actually, an increase of about 20% in micropore volume was observed. The surface properties of the added particles, their hydrophilicity for a surface covered with hydroxyl groups and hydrophobicity for a surface covered with methyl or octyl groups, did not affect the character of porosity. Therefore, it can be assumed that there was no formation of chemical bonds between the particle surface and the surrounding gel because, if chemical bonds were formed, there would be a difference.

When an organic amine, a substance with a higher basicity than dibutyltin dilaurate but a lower basicity than inorganic bases (e.g., ammonia or hydroxides of alkaline metals), was used as the catalyst, important changes in porosity occurred [21]. The formation of micropores was substantially suppressed and a considerable amount of mesopores were formed, their width achieving 5 nm depending on the type and concentration of organic amine used. Due to the structure of their molecules, some primary amines with a longer alkyl chain (such as octyl or dodecylamine) might function as non-ionic surfactants that reduce the surface tension.

The texture of SiGel-ZnO-d/oa was more complex, exhibiting both micro- and mesoporosity, due to the combination of both dibutyltin dilaurate and organic amine catalysts. The commercial KSE OH consolidant exhibited similar texture properties, but SP FX was practically non-porous.

Thus, to suppress the formation of micropores, which have a negative effect on the cracking of gels due to their high capillary pressure [2] and to induce the formation of advantageous, substantially wider mesopores, an organic amine is preferable as a catalyst [21].

3.2.2. Mechanical Properties of the Xerogels in Microscale

First, the elastic character of the xerogels was confirmed by the frequency sweep—constant force measurements—which showed that their storage modulus did not depend on the frequency (see Figure 1a).

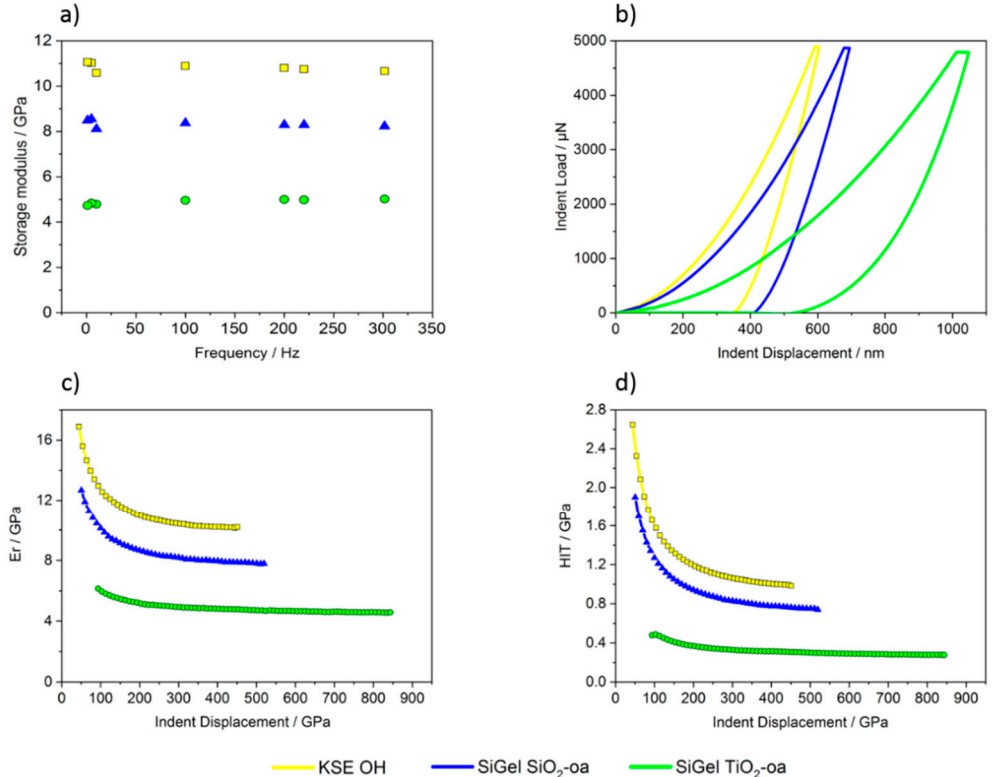

**Figure 1.** Mechanical properties of the xerogel on a microscale level. (**a**) Variation of the storage modulus with the sweep frequency; (**b**) load-displacement curves; (**c**) Dependence of Young's modulus (**c**) and hardness (**d**) on indent displacement.

The mechanical properties in microscale of the xerogel were determined by a nanoindentation testing. Using the stress–strain curve, the contact stiffness and reduced elastic modulus, representing the Young's modulus of elasticity and indentation hardness, respectively, were obtained. The example of the nanoindentation curves is shown in Figure 1b.

The catalyst used decisively affected the mechanical performance of the xerogels (Table 4). Both reference xerogels SiGel-d and KSE OH, for which an organometallic catalyst was used, exhibited much higher hardness than those catalyzed by n-octylamine. The indentation hardness correlated with the Young's modulus and the contact depth. The shape of the nanoindentation curves for the harder xerogels exhibited the character of glass-like deformation, while that for softer xerogels was polymer-like. The difference in the mechanical properties is probably due to the compact structure containing only narrow micropores of the xerogels catalyzed by organometallic compounds. The xerogels prepared using an n-octylamine catalyst showed lower values of both mechanical parameters, which is connected with their more open mesoporous structure.

**Table 4.** Mechanical properties of the xerogels determined by nanoindentation.

| Xerogel | Young's Modulus/GPa | Hardness/GPa | Contact Depth/nm |
| --- | --- | --- | --- |
| SiGel-d | 7.74 ± 0.52 | 1.04 ± 0.09 | 182.21 ± 9.35 |
| SiGel-oa | 4.49 ± 0.14 | 0.41 ± 0.02 | 297.88 ± 6.50 |
| SiGel-SiO$_2$-methyl-oa | 3.83 ± 0.06 | 0.45 ± 0.01 | 285.80 ± 4.43 |
| SiGel-TiO$_2$-oa | 4.72 ± 0.03 | 0.33 ± 0.00 | 349.96 ± 2.10 |
| KSE OH | 11.44 ± 0.25 | 1.32 ± 0.05 | 175.75 ± 4.34 |

Generally, the hardness and elasticity were depth dependent (Figure 1c,d); those of the outermost surface layers being higher compared to the bulk, probably due to the faster hydrolysis and condensation of the ethyl silicate species within the surface layers of the xerogel block. However, from the consolidation viewpoint, the consolidation should be homogeneous within the stone profile. This feature was reasonably fulfilled using our consolidants, better than with the commercial ones. The storage modules determined by this method are in agreement with those in Figure 1a.

### 3.2.3. Accelerated Weathering of Xerogels

The test showed that the durability of the xerogels substantially depended on the nanoparticles used as additives. The cohesion of weathered xerogels prepared using different nanoparticles, the formulation being otherwise the same, differed considerably (Figure 2). The SiGel-SiO$_2$-methyl-oa xerogel exhibited excellent stability, while SiGel-SiO$_2$-oa and SiGel-ZnO-oa/d performed much worse. The sorption measurements showed that, after the accelerated weathering test, the structure of the tested xerogels significantly changed. The structure of SiGel-SiO$_2$-methyl-oa and SiGel-SiO$_2$-octyl-oa xerogel was more stable, however, a decrease in surface area and pore volume was observed. This process is similar to the Oswald ripening, during which destruction of narrow pores and formation of rigid structure occur. The structure of SiGel-SiO$_2$-oa significantly disintegrated, thus, the surface area increased. Xerogel SiGel-ZnO-oa/d is anomalous due to sedimentation of the ZnO nanoparticles during the sol–gel process, which fundamentally affected the structure of the material. Therefore, a high increase of the surface area and pore volume was observed. Compared to the macroscopic material block, for a thin layer to form in narrow pores of the rock, it can be expected that this process will be less pronounced. Thus, it does not mean that ZnO nanoparticles are unsuitable as additives for consolidants.

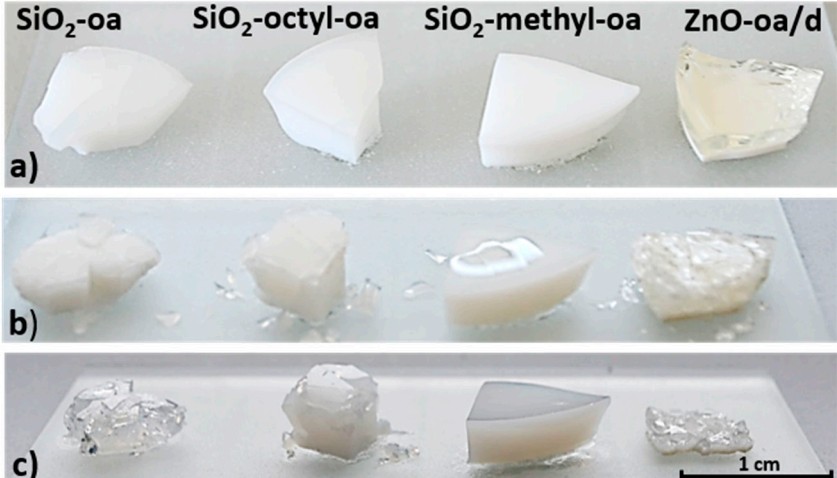

**Figure 2.** Effect of weathering on the consistency of the xerogels. (**a**) fresh; (**b**) after one WAT cycle; (**c**) after two WAT cycles.

### 3.3. Application of the Consolidants on the Sandstone

#### 3.3.1. Uptake of the Consolidants

When applied on the sandstone, the uptake of all the tested consolidants by the capillary suction was similar to about 10 wt.% (Table 5). However, their dry matter yield differed substantially.

**Table 5.** Sandstone uptake by capillary suction.

| Sandstone Treatment | Consolidant Uptake/ wt.% | Dry Matter Yield/ wt.% | Water Uptake/ wt.% |
|---|---|---|---|
| Untreated | - | - | 11.60 ± 0.5 |
| SiGel-oa | 10.1 ± 0.5 | 4.0 ± 0.2 | 0.09 ± 0.01 |
| SiGel-SiO$_2$-oa | 8.0 ± 0.4 | 4.3 ± 0.2 | 0.06 ± 0.01 |
| SiGel-SiO$_2$-methyl-oa | 10.3 ± 0.5 | 4.2 ± 0.3 | 0.12 ± 0.02 |
| SiGel-SiO$_2$-octyl-oa | 9.2 ± 0.4 | 4.8 ± 0.3 | 0.12 ± 0.01 |
| SiGel-TiO$_2$-oa | 10.1 ± 0.6 | 4.8 ± 0.4 | 0.11 ± 0.01 |
| SiGel-ZnO-oa/d | 10.2 ± 0.6 | 3.7 ± 0.2 | 0.32 ± 0.02 |
| HAP | 13.9 ± 0.7 | 0.24 ± 0.03 | 9.8 ± 0.5 |
| HAP+Ca(OH)$_2$ | 10.8 ± 0.5 | 0.31 ± 0.03 | 9.8 ± 0.6 |
| KSE OH | 10.6 ± 0.4 | 2.7 ± 0.11 | 2.3 ± 0.1 |
| SP FX | 9.6 ± 0.4 | 1.1 ± 0 05 | 9.5 ± 0.5 |

Regarding octylamine-catalyzed SiGel consolidants, it was in a narrow range of about 4–5 wt.% Concerning the commercial sample SP FX, this yield was only 1.1 wt.%, which roughly corresponds to the lower content of TEOS in this consolidant (see Table 1). Regarding the other commercial product KSE OH, which contains a higher percentage of oligomeric TEOS (75%) than the SiGel samples (50%), the yield was surprisingly only 2.7 wt.%. The reasons for such a low value may be the different solvent used, the presence of nanoparticles and, especially, the catalyst, which controls the rate of hydrolysis and polycondensation. The insufficient rate of these processes may lead to a loss of the active species (alkoxide) through faster evaporation.

Another marked difference between the octylamine-catalyzed SiGel consolidants and the reference KSE OH was in the water uptake. That of the former group of consolidants was smaller by an order of magnitude. The explanation for the smaller uptake by SiGel consolidants seems to be their higher hydrophobicity caused by the use of the amine catalyst. The high water uptake of SP FX and HAP consolidants, approaching that of the untreated rock, is clearly due to a rather low dry matter yield.

### 3.3.2. Mechanical Testing in Microscale

The direct determination of the location of the consolidant xerogel within the porous system of the sandstone and the measuring of its mechanical properties was carried out by nanoindentation. The consolidant was applied on the stone by capillary suction. To localize the consolidant, a series of indents (applied stress of 1 mN) were performed to identify a xerogel deformation pattern. The contact depth for the mineral grains in the sandstone, SiGel-d consolidant and a soft epoxy resin was ca. 100, 180 and 500 nm, respectively.

On the stone, The SiGel-d xerogel formed a thin layer, which corresponded to the Young's modulus and hardness of 11.40 GPa and 0.93 GPa, respectively. Furthermore, the xerogel penetrated through the porous system and was detected in the lowest point of 700 μm, in which the Young's modulus and hardness was 10.97 and 0.74 GPa, respectively. Regarding the SiGel-oa, the xerogel was not observed on the sandstone surface. First deposits of the xerogel were located in the depth of 250 μm. Concerning this xerogel, the contact depth was ca. 300 nm, which was higher compared to the SiGel-d. The probable explanation is the better penetration of the SiGel-oa consolidant and the higher compliance of the xerogel formed due to the presence of wider mesopores in comparison with the narrower micropores of the SiGel-d xerogel.

Compared to the pristine SiGel-oa and SiGel-d xerogels, their counterparts localized in the pores of the stone exhibited a roughly two-fold increase in the Young's modulus. This difference may be due to a confinement effect of the pore walls [22–26], which may increase the Young's modulus owing to the wrapping of the gel by hard stone material. Considering the hardness, however, no such increase was observed. This observation implies that the microstructure of the xerogel was not influenced by

the presence of pore walls, which is a reasonable conclusion provided there was no formation of the chemical bonds between the gel and the surface of the pore walls.

### 3.3.3. Mechanical Testing in Macroscale

Due to the treatment with octylamine-catalyzed SiGel consolidants, there is a systematic decrease in the surface area (Table 6), which is in agreement with the smooth and uncracked character of the xerogels. Sample SiGel ZnO-oa/d exhibited an anomalous behaviour with a ten-fold increase in the surface area and formation of microporosity. The surface of the KSE OH-treated sample is comparable with the untreated sample, even if it should decrease due to the filling of pores, especially of the finer ones. The reasonable explanation is a cracking of the xerogel inside the pores, which created an additional surface area. The performance of the SP FX consolidant, characterized by an increase in the surface area and formation of micropores, is comparable with the SiGel ZnO-oa/d. Using the DAP consolidants, there was only a limited decrease in the surface area, which is in agreement with the very low dry matter yield for these samples.

**Table 6.** Properties of sandstone treated with various consolidants weathered by 50 cycles (see Table 2).

| Sandstone Treatment | BET Area Fresh [1]/ $m^2\,g^{-1}$ | BET Area Aged [2]/ $m^2\,g^{-1}$ | DRMS Fresh [3]/ N | DRMS Aged [4]/ N | Δ DRMS [5]/ % |
|---|---|---|---|---|---|
| Untreated | 0.56 | - | 9.7 ± 0.5 | 3.2 ± 0.2 | −67 |
| SiGel-oa | 0.05 | 5.90 | 11.6 ± 0.8 | 9.2 ± 1.2 | −21 |
| SiGel SiO$_2$-oa | 0.05 | 16.77 | 15.5 ± 0.8 | 12.1 ± 0.8 | −22 |
| SiGel SiO$_2$-octyl-oa | 0.15 | 8.41 | 10.5 ± 0.7 | 4.0 ± 1.0 | −62 |
| SiGel SiO$_2$-methyl-oa | 0.06 | 7.42 | 17.3 ± 1.9 | 12.7 ± 1.0 | −27 |
| SiGel TiO$_2$-oa | 0.02 | 0.17 | 23.3 ± 2.5 | 18.1 ± 1.4 | −22 |
| SiGel ZnO-oa/d | 5.10 * | 10.60 * | 54.9 ± 1.7 | 60.9 ± 2.2 | +11 |
| KSE OH | 0.46 | 2.54 | 25.2 ± 1.4 | 18.5 ± 1.5 | −27 |
| SP FX | 2.10 * | 3.00 | 22.7 ± 1.6 | 16.4 ± 1.4 | −28 |
| HAP | 0.23 | 0.42 | 18.5 ± 1.3 | 18.1 ± 1.6 | −2 |
| HAP+Ca(OH)$_2$ | 0.23 | 0.73 | 17.8 ± 1.4 | 17.6 ± 1.5 | −1 |

* Samples containing micropores; [1] BET surface area before weathering; [2] BET surface area after weathering; [3] Drilling resistance force before weathering; [4] Drilling resistance force after weathering; [5] Change in the Drilling resistance force due to weathering.

Compared to the untreated sandstone, the drilling resistance for the treated samples increased (Table 6). However, this increase depended substantially on the consolidant used.

Regarding the consolidant SiGel-oa without nanoparticles, only a very limited increase in the force was observed. However, importantly, the stone retained its cohesion after weathering, as the resistance force decreased only by 21%. Concerning the SiGels containing SiO$_2$ nanoparticles, some modification effect of their surface was observed. The surface coating with octyl-groups decreased the resistance force in comparison with the other two types of particles. Moreover, SiGel SiO$_2$-octyl-oa exhibited low durability. The reason is probably the practically complete coverage of the SiO$_2$ particles with octyl-groups. The performance of SiGel containing TiO$_2$ nanoparticles was slightly better than that of both commercial products, exhibiting higher durability. Interestingly, the SiGel containing ZnO nanoparticles, catalyzed by both amine and organometallic compounds, exhibited an exceptionally high resistance force, even after the weathering treatment. This performance might be due to the unusual micro-mesoporous texture of the xerogel and the solubility of the ZnO nanoparticles. A reaction occurred between the zinc ions and the xerogel-formed ions during the course of the sol-gel process. Similar reactions can be expected during the process of accelerated weathering. Finally, both HAP samples exhibited a reasonable degree of consolidation and excellent durability.

The BET surface areas measured after the weathering treatment partially correlated with the resistance force. Samples with a small surface area after the weathering treatment (less than 1 m$^2$

$g^{-1}$), i.e., SiGel TiO$_2$-oa, HAP and HAP+Ca(OH)$_2$, exhibited high durability. The large surface area of the SiGel-oa sample and those containing SiO$_2$ nanoparticles correlated with their lower durability, with the exception of SiGel SiO$_2$-oa. SiGel ZnO-oa/d exhibited unusual surface areas, which might be connected with the structural changes, as suggested above.

## 4. Conclusions

This comparative study shows that our suggested approaches enabled us to obtain consolidants whose performance surmounts that of commercial ones serving as a reference. We evaluated the durability of the consolidation effect after accelerated weathering as a difference in the surface area and a change in the drilling resistance force. Our alkoxysilane-based PMC (ZnO, TiO$_2$) consolidants exhibited good mechanical performance with excellent durability. The performance of the consolidant containing ZnO nanoparticles, catalyzed by a mixture of amine and organometallic compounds, was exceptional, exhibiting unusually high strength and durability. Moreover, the TiO$_2$ nanoparticles used exhibited negligible toxicity toward human lung cells [11]. Concerning the diammonium hydrogen phosphate-based materials already used in the high-performance consolidation of carbonate materials, our study shows their applicability to the consolidation of sandstone. These materials formed a thin compact layer covering the inner surface of the stone; this layer survived the weathering treatment without any crack formation. Regarding our future work, we intend to modify these highly promising diammonium hydrogen phosphate-based materials by the addition of TiO$_2$ and/or ZnO nanoparticles, and to explore their consolidation performance and associated additional biocidal and self-cleaning functionality, especially in relation to problematic calcareous sandstone or sandy limestone.

**Author Contributions:** Conceptualization, J.R.; methodology, J.R., R.Z., M.R., J.L.; investigation, J.R., R.Z., M.R., J.L.; data curation, J.R., R.Z., M.R.; writing—Original draft preparation, J.R., R.Z.; writing—Review and editing, J.R., R.Z.; supervision, J.R.; project administration, J.R.; funding acquisition, J.R.

**Funding:** This research was funded by the Czech Science Foundation (GACR, Grant No. 17-18972S).

**Acknowledgments:** This work was supported by the Ministry of Education, Youth and Sports of the Czech Republic and The European Union—European Structural and Investments Funds in the frame of Operational Programme Research, Development and Education—Project Pro-NanoEnviCz (Project No. CZ.02.1.01/0.0/0.0/16_013/0001821), which provided an access to the Micromeritics 3Flex apparatus.

**Conflicts of Interest:** The authors declare no conflict of interest.

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
