# Peer review of "Potential of Advanced Consolidants for the Application on Sandstone"

_applsci, doi:10.3390/app9235252_

Round 1

Reviewer 1 Report

The article contains important information and data that will contribute to the body of knowledge on the consolidation of stone.

The presentation of methods and data is in some cases incomplete and in other cases unclear.
Here are specific comments:

The introduction is rather poor both in the identification of the need for consolidation, and for the state of the art about modified TEOS and nanocomposites used for consolidation purposes. A weak or critical aspect is the choice of the experimental set-up using a sandstone (high porosity) as quarried, i.e. without any deterioration or ageing procedure before treatment; AA should justify this choice. In order to better understand the characteristics of the lithotype employed in this research the authors should provide pore size distribution, specific pore volume, average pore radius and apparent density. The authors should report the dimensions of the specimens for the consolidation treatments. Also, the number of the specimens per each treatment. The AA should report the conditions (T and RH%) during treatment and the curing conditions (T, RH% and time of curing) where the specimens were stored. The authors should furnish along with the average values of all data as well as standard deviations. Table 4: the mechanical properties of some xerogels are missing. The authors should refer the standards used to perform the measurements, i.e. for water uptake, etc. Water uptake: was residual hydrophobicity removed from the TEOS treatments before testing?

In general, the authors should present the design performance and evaluation of the tests in a more comprehensive framework where the conclusions should be supported from the data.

Author Response

The article contains important information and data that will contribute to the body of knowledge on the consolidation of stone. The presentation of methods and data is in some cases incomplete and in other cases unclear.

Here are specific comments:

1.The introduction is rather poor both in the identification of the need for consolidation, and for the state of the art about modified TEOS and nanocomposites used for consolidation purposes.

We have completely rewritten the Introduction, which now reads:

Alkoxysilanes have been the most widely used stone consolidants over the past forty years [1]. Two compounds have been dominant: methytrimethoxysilane (MTMOS) and tetraethoxysilane (TEOS). A number of products based on TEOS and its oligomers are available on the market. Their advantages are well known: their versatility, chemical stability, low viscosity, the ability to form silicon-oxygen-silicon (Si-O-Si) bonds [2], harmlessness of the compounds (e.g. water, ethanol) formed during gelation, avoidance of over-consolidation of the stone surface, and, last but not least, cost effectiveness and ease of use. Silicon-oxygen bonds are contained in many minerals, thereby contributing to the compatibility between stone and TEOS-based alkoxysilanes. Owing to the advantages listed above, such consolidants easily penetrate deeply into the stone porous system [3]. They can replace an original binder lost during weathering and, thereby, significantly enhance the cohesion of the stone grains and, thus, overall structure [1].

However, there are serious drawback concerning the application of TEOS-based consolidants. In particular, the resulting silica gel shrinks and tends to develop cracks inside the stone due to the stress caused by increasing capillary force during aging and drying [1,3]. This particular situation leads to the formation of brittle gel fragments that create a secondary capillary network in the stone, which can accelerate the rate of deterioration. As the xerogel formed by the gelation of TEOS and oligomeric ethylsilicates are compact with only very narrow micropores (1–2 nm in width), the water vapor permeability is substantially hindered, which represents a major danger for the stability of the treated stone. Furthermore, the physico-chemical and structure properties of these consolidant xerogels differed from those of nature sandstones, which consist of quartz particles and inorganic binder.

This issue has been of major research interest in stone consolidation applications over the past twenty years. There are several approaches to overcoming the above-mentioned shortcomings. Blanco et al. showed that by adding drying control chemical additives (DCCA), such as formamide, the surface tension of the sol was changed and, thus, the evaporation rate was reduced, leading to the suppression of crack formation [4]. Mosquera et al. also reduced cracking by the addition of suitable surfactants (alkylamines), which decreased capillary pressure and surface tension [5]. Another strategy involves the incorporation of elastic segments, such as hydroxyl-terminated polydimethylsiloxane (PDMS), into the xerogel structure; these segments effectively removed the stress caused by capillary pressure [5–9]. Another more general approach is based on the incorporation of various oxide nanoparticles, such as SiO2, TiO2, Al2O3 and ZnO (so-called particle-modified consolidants, PMC) [3,8,10–15]. Due to the increase in the pore size of PMC-based xerogel and the improvement of its mechanical properties, cracking was reduced and the consolidation effect enhanced. However, while they have shown significant improvements, these ´new´ formulations exhibit considerable limitations due to the complexity of the treated stone objects.

In the development of any novel consolidant, the selection of a suitable stone substrate is fundamental. In principle, three main substrates can be used: (i) a naturally aged substrate, (ii) an artificially aged substrate, and (iii) stone as quarried [15]. The first option provides the advantage of weathering under real world conditions and potentially reliable technological transfer into conservation practice. However, there are issues, namely the insufficient availability of stone samples for extensive laboratory testing and the natural inhomogeneity. The modification of stone by artificial weathering (second option) is rather more problematic as the aging processes differ considerably from those in nature. The oft-used thermal treatment, especially of sandstone, causes physico-chemical changes, such as binder hardening, that render such treated materials unsuitable for consolidation tests. The freeze-thaw cycles are unsuitable for large-scale testing. In addition, salt crystallization and/or acid attack make it difficult to interpret consolidation performance. In terms of quarried stone, the test material must be very carefully selected so that it accurately mimic real weathered one.

The Czech Republic is particularly suitable in this respect because it is rich in the sandstone quarries, meaning that a wide range of samples differing in texture and mechanical properties are available. Our previous studies have shown that sandstone from the Msene quarry in Central Bohemia is suitable for the evaluation of consolidation performance because it is characterized by high porosity and relatively low mechanical strength [[11,13,14]]. This sandstone is characterized by high purity (over 95% of quartz and small amount of binder), homogeneity and well-defined monomodal porous system with a narrow pore size distribution. Moreover, this stone was used as a construction material for extremely important historical monuments in Prague, such as St. Vitus´s Cathedral, Charles Bridge, the Old Town Hall and several government buildings.

The third fundamental aspect, besides the selection of stone substrate to be tested and the choice of consolidant, is the methodology how the success of consolidation will be evaluated. From the practical point of view, the long-term durability of the consolidation effect is decisive than the initial level of consolidation. If there was a fast loss of the consolidation effect, such consolidation technology would not be suitable for application.

Therefore, in this study, we devoted to the aspect of durability special attention and used it as a criterion for assessment of the consolidation outcome. Specifically, we consolidated and artificially weathered Czech-quarried sandstone. We assessed the suitability of various consolidant formulations, including two TEOS-based commercial products (Remmers KSE OH, SurfaPore FX SB), a self-developed TEOS-based PMC consolidant [11], and two self-developed phosphate-based consolidants. To determine the durability of each type of sandstone consolidation, mechanical tests were performed on a micro- (nanoindentation) and macroscale (drilling resistance measurements). To evaluate the cohesion of the consolidant xerogels, both before and after incorporation into the sandstone pores, sorption experiments were carried out in the gas phase.

2.A weak or critical aspect is the choice of the experimental set-up using a sandstone (high porosity) as quarried, i.e. without any deterioration or ageing procedure before treatment; AA should justify this choice.

We have justify the choice of the stone substrate used for testing in the following paragraphs in the Introduction:

In the development of any novel consolidant, the selection of a suitable stone substrate is fundamental. In principle, three main substrates can be used: (i) a naturally aged substrate, (ii) an artificially aged substrate, and (iii) stone as quarried [15]. The first option provides the advantage of weathering under real world conditions and potentially reliable technological transfer into conservation practice. However, there are issues, namely the insufficient availability of stone samples for extensive laboratory testing and the natural inhomogeneity. The modification of stone by artificial weathering (second option) is rather more problematic as the aging processes differ considerably from those in nature. The oft-used thermal treatment, especially of sandstone, causes physico-chemical changes, such as binder hardening, that render such treated materials unsuitable for consolidation tests. The freeze-thaw cycles are unsuitable for large-scale testing. In addition, salt crystallization and/or acid attack make it difficult to interpret consolidation performance. In terms of quarried stone, the test material must be very carefully selected so that it accurately mimic real weathered one.

The Czech Republic is particularly suitable in this respect because it is rich in the sandstone quarries, meaning that a wide range of samples differing in texture and mechanical properties are available. Our previous studies have shown that sandstone from the Msene quarry in Central Bohemia is suitable for the evaluation of consolidation performance because it is characterized by high porosity and relatively low mechanical strength [[11,13,14]]. This sandstone is characterized by high purity (over 95% of quartz and small amount of binder), homogeneity and well-defined monomodal porous system with a narrow pore size distribution. Moreover, this stone was used as a construction material for extremely important historical monuments in Prague, such as St. Vitus´s Cathedral, Charles Bridge, the Old Town Hall and several government buildings.

3.In order to better understand the characteristics of the lithotype employed in this research the authors should provide pore size distribution, specific pore volume, average pore radius and apparent density.

The required date were added to the manuscript (part 3.1)

4.The authors should report the dimensions of the specimens for the consolidation treatments. Also, the number of the specimens per each treatment.

The size of the block was 30 mm x 30 mm x 30 mm. The experiments were done in triplicate.

5.The AA should report the conditions (T and RH%) during treatment and the curing conditions (T, RH% and time of curing) where the specimens were stored.

During the treatment and curing, the conditions were controlled by a climatic chamber at the temperature of 25°C and RH of 75%.

6.The authors should furnish along with the average values of all data as well as standard deviations.

We have added the average values and their standard deviations where available.

7.Table 4: the mechanical properties of some xerogels are missing.

The nanoindentation data were obtained only for given xerogels because of the limited access to this experimental technique.

8.The authors should refer the standards used to perform the measurements, i.e. for water uptake, etc.

As reference the two commercial products served as a reference for comporasion with the self-developed consolidants.

Water uptake: was residual hydrophobicity removed from the TEOS treatments before testing?

For samples treated with ethylsilicate-containing consolidant, the hydrophobicity was not removed. The consolidated sandstone specimens were cured at 25 °C and 75 % RH for six months.

In general, the authors should present the design performance and evaluation of the tests in a more comprehensive framework where the conclusions should be supported from the data.

We have completely rewritten the Introduction. We precisely explain the reason for the selection of the test stone substrate and the use of methodology approach, explained in the Introduction section ( see answer to the 1st comment).

Reviewer 2 Report

Manuscript Number: applsci-645336

The manuscript by Remzova et al  entitled, “Potential of Advanced Consolidants for the  Application on Sandstone Rocks” intends to provide a detailed look of the efficiency and durability of consolidants (commercial and self-developed ones). The authors used different techniques such as  nitrogen adsorption, mercury intrusion, nanoindentation and drilling resistance measurement. Stone samples were evaluated after application of the consolidants and after artificial ageing tests.

The topic of the research is important for Cultural Heritage not only in R&I but also relevant for those who work/decide in built heritage, namely conservators-restorers working with this particular type of problem (loss of cohesion of stone materials) and decision markers.

The structure of the article is clear and methodology and results are n well stated.  The bibliographical references are current and pertinent for presentation of the problem/situation and discussion of the results.

The article meets the requirements to consider its publication in the journal. The research is interesting and fits the scopus of the journal.  Nevertheless, it does not meet enough quality to be published without going through a MINOR revision. There are several questions and aspects that must be clarified.

In the pdf document I introduced my comments/questions.

Author Response

The manuscript by Remzova et al  entitled, “Potential of Advanced Consolidants for the  Application on Sandstone Rocks” intends to provide a detailed look of the efficiency and durability of consolidants (commercial and self-developed ones). The authors used different techniques such as  nitrogen adsorption, mercury intrusion, nanoindentation and drilling resistance measurement. Stone samples were evaluated after application of the consolidants and after artificial ageing tests.

The topic of the research is important for Cultural Heritage not only in R&I but also relevant for those who work/decide in built heritage, namely conservators-restorers working with this particular type of problem (loss of cohesion of stone materials) and decision markers.

The structure of the article is clear and methodology and results are n well stated.  The bibliographical references are current and pertinent for presentation of the problem/situation and discussion of the results.

The article meets the requirements to consider its publication in the journal. The research is interesting and fits the scopus of the journal. 

Nevertheless, it does not meet enough quality to be published without going through a MINOR revision. There are several questions and aspects that must be clarified.

In the pdf document I introduced my comments/questions.

The authors thank the reviewer for careful revision of the manuscript and a number of detailed remarks. We have accepted all of them and changed the manuscript accordingly. We have also completely rewritten the Introduction in order to explain our approach in a clearer way.

Reviewer 3 Report

This paper is well written and for this reason I suggest to be published

Author Response

This paper is well written and for this reason I suggest to be published.

The authors thank the reviewer for support of our manuscript and do hope that will be of interest for broader readership.

Round 2

Reviewer 1 Report

The review of the article is positive and its acceptance proceeds.